# GENESPACE tracks regions of interest and gene copy number variation across multiple genomes

**John T Lovell[1,2]\***, **Avinash Sreedasyam[1]**, **M Eric Schranz[3]**, **Melissa Wilson[4]**, **Joseph W Carlson[2]**, **Alex Harkess[1,5]**, **David Emms[6]**, **David M Goodstein[2]**, **Jeremy Schmutz[1,2]**

[1]Genome Sequencing Center, HudsonAlpha Institute for Biotechnology, Huntsville, United States; [2]Joint Genome Institute, Lawrence Berkeley National Laboratory, Berkeley, United States; [3]Biosystematics Group, Wageningen University and Research, Wageningen, Netherlands; [4]Center for Evolution and Medicine, School of Life Sciences, Arizona State University, Tempe, United States; [5]Department of Crop, Soil, and Environmental Sciences, Auburn University, Auburn, United States; [6]Oxford University, Oxford, United Kingdom

**Abstract** The development of multiple chromosome-scale reference genome sequences in many taxonomic groups has yielded a high-resolution view of the patterns and processes of molecular evolution. Nonetheless, leveraging information across multiple genomes remains a significant challenge in nearly all eukaryotic systems. These challenges range from studying the evolution of chromosome structure, to finding candidate genes for quantitative trait loci, to testing hypotheses about speciation and adaptation. Here, we present GENESPACE, which addresses these challenges by integrating conserved gene order and orthology to define the expected physical position of all genes across multiple genomes. We demonstrate this utility by dissecting presence–absence, copy-number, and structural variation at three levels of biological organization: spanning 300 million years of vertebrate sex chromosome evolution, across the diversity of the Poaceae (grass) plant family, and among 26 maize cultivars. The methods to build and visualize syntenic orthology in the GENESPACE R package offer a significant addition to existing gene family and synteny programs, especially in polyploid, outbred, and other complex genomes.

**\*For correspondence:**
jlovell@hudsonalpha.org

**Competing interest:** The authors declare that no competing interests exist.

## Editor's evaluation

GENESPACE is a new and straightforward computational tool to include synteny information in the calculation of genome-wide sets of orthologs. The development of this tool is very timely as more and more complete chromosome-scale assembled genomes are becoming available. While the assembly problem has been solved, this is not the case for multiple genome comparisons, and GENESPACE is an important step to help remedy this gap in our comparative genomics toolbox.

## Introduction

De novo genome assemblies and gene model annotations represent increasingly common resources that describe the sequence and positions of protein coding and intergenic regions within a single genotype. Evolutionary relationships among these DNA sequences form the foundation of many molecular tools in modern medical, breeding, and evolutionary biology research.

**eLife digest** The genome is the complete DNA sequence of an individual. It is a crucial foundation for many studies in medicine, agriculture, and conservation biology. Advances in genetics have made it possible to rapidly sequence, or read out, the genome of many organisms. For closely related species, scientists can then do detailed comparisons, revealing similar genes with a shared past or a common role, but comparing more distantly related organisms remains difficult.

One major challenge is that genes are often lost or duplicated over evolutionary time. One way to be more confident is to look at 'synteny', or how genes are organized or ordered within the genome. In some groups of species, synteny persists across millions of years of evolution. Combining sequence similarity with gene order could make comparisons between distantly related species more robust.

To do this, Lovell et al. developed GENESPACE, a software that links similarities between DNA sequences to the order of genes in a genome. This allows researchers to visualize and explore related DNA sequences and determine whether genes have been lost or duplicated. To demonstrate the value of GENESPACE, Lovell et al. explored evolution in vertebrates and flowering plants. The software was able to highlight the shared sequences between unique sex chromosomes in birds and mammals, and it was able to track the positions of genes important in the evolution of grass crops including maize, wheat, and rice.

Exploring the genetic code in this way could lead to a better understanding of the evolution of important sections of the genome. It might also allow scientists to find target genes for applications like crop improvement. Lovell et al. have designed the GENESPACE software to be easy for other scientists to use, allowing them to make graphics and perform analyses with few programming skills.

Perhaps the most crucial inference to make when comparing genomes revolves around homologous genes, which share an evolutionary common ancestor and ensuing sequence or protein structure similarity. Analyses of homologs, including comparative gene expression, epigenetics, and sequence evolution, require the distinction between orthologs, which arise from speciation events, and paralogs, which arise from sequence duplications (*see Box 1 for definitions*). In some systems, this is a simple task where most genes are single copy, and orthologs are synonymous with reciprocal best-scoring protein BLAST hits. Other sequence similarity approaches such as OrthoFinder (*Emms and Kelly, 2019*; *Emms and Kelly, 2015*) leverage graphs and gene trees to test for orthology, permitting more robust analyses in systems with gene copy number (CNV) or presence–absence variation (PAV). However, whole-genome duplications (WGDs), chromosomal deletions, and variable rates of sequence evolution, such as subgenome dominance in polyploids, can confound the evidence of orthology from sequence similarity alone.

The physical position of homologs offers a second line of evidence that can help overcome challenges posed by WGDs, tandem arrays, heterozygous-duplicated regions, and other genomic complexities (*Drillon et al., 2020*; *Haug-Baltzell et al., 2017*; *Wang et al., 2012*). Synteny, or the conserved order of DNA sequences among chromosomes that share a common ancestor, is a typical feature of eukaryotic genomes. In some taxa, synteny is preserved across hundreds of millions of years of evolution and is retained over multiple WGDs (*Jiao et al., 2014*; *Simakov et al., 2020*; *Zhao and Schranz, 2019*). Like chromosome-scale synteny, conserved gene order collinearity along local regions of chromosomes can provide evidence of homology, and in some cases enable determination of whether two regions diverged as a result of speciation or a large-scale duplication event (*Drillon et al., 2020*). Combined, evidence of gene collinearity and sequence similarity should improve the ability to classify paralogous and orthologous relationships beyond either approach in isolation.

Integrating synteny and collinearity into comparative genomics pipelines also physically anchors the positions of related gene sequences onto the assemblies of each genome. For example, by exploring only syntenic orthologs it is possible to examine all putatively functional variants within a genomic region of interest, even those in genes that are absent in the focal reference genome (*Lovell et al., 2018*). Such a 'pan-genome annotation' framework (*Lovell et al., 2021a*) permits access to multi-genome networks of high-confidence orthologs and paralogs, regardless of ploidy or other complicating aspects of genome biology. Here, we present GENESPACE, an analytical pipeline that explicitly links synteny and sequence similarity to provide high-confidence inference about networks of genes

## Box 1. Definitions

**Orthogroup** — a set of genes across multiple genomes derived from a single ancestral gene.
**Ortholog** — a pair of orthogroup members in two species derived from a single gene in their most recent common ancestor.
**Paralog** — orthogroup members derived from a duplication event since speciation.
**Homeolog** — paralogs derived from a whole-genome duplication.
**Tandem array** — paralogs in proximity on a chromosome within a genome.
**Gene collinearity** — retained order of genes across species due to common ancestry.
**Synteny** — like collinearity but at larger scales, like chromosome arms.
**Pan-genome annotation** — A set of orthogroups across multiple genomes, placed along the coordinate system of a specified reference genome.

that share a common ancestor and represents these networks as a 'pan-genome annotation'. We then leverage this framework to explore gene family evolution in flowering plants, mammals, and reptiles.

## Results and discussion

### GENESPACE syntenic orthology methods to compare multiple complex genomes

Comparative genomics across the complex evolutionary histories of eukaryotes typically requires equally varied input and analytical pipelines depending on researchers' goals and study systems. For example, synteny between closely related haploid assemblies is often inferred by exploring only 1:1 reciprocal best-scoring hits with MCScanX (*Wang et al., 2012*). Alternatively, polyploid genomes are typically split into subgenomes so that homeologs are viewed by clustering algorithms like OrthoFinder (*Emms and Kelly, 2019*; *Emms and Kelly, 2015*) as orthologous and not paralogous. While expert knowledge that informs these analytical decisions can dramatically improve precision, this knowledge is not available in many systems. These issues boil down to a simple circular problem: a priori knowledge of gene copy number is needed to effectively infer orthology and synteny, yet measures of synteny and orthology are needed to infer copy number between a pair of sequences.

GENESPACE resolves this circular problem by operating on a foundational assumption: homeologs should be exactly single copy within any syntenic region between a pair of genomes. There are two major violations of this assumption that cause copy number variation within a syntenic block: (1) tandem arrays and (2) gene PAV. GENESPACE addresses these complexities directly in two ways. First, physically proximate multigene families (hereon, 'tandem arrays', *Box 1*) are condensed to the physically most central gene of the array. Gene rank order along the genome is recalculated on these 'array representative' genes, effectively masking copy number variation due to tandem arrays. Second, synteny is inferred in a pairwise manner only using 'potential anchor' protein BLAST hits (hereon 'hits') where both the query and target genes are in the same orthogroup. The rank-order positions of these potential anchor genes are also recalculated prior to synteny inference, which effectively masks orthogroups missing a gene in one genome (i.e., PAV). Thus, GENESPACE operates on OrthoFinder-derived orthogroups with one-to-one relationships for any accurately defined syntenic region, regardless of ploidy or level of sequence conservation.

Given GENESPACE's reliance on syntenic regions between genomes, errors in syntenic block coordinates can have major effects on downstream estimates. Therefore, we crafted a sensitive pipeline to infer syntenic regions (see *pipeline overview in Figure 1*). To demonstrate its functionality, we ran GENESPACE on seven pairs of genomes spanning closely related maize cultivars to ancient mammal–bird divergence (~300 M ya; *Table 1*, see Materials and methods). The three comparisons between vertebrate genomes have no recent history of WGDs, while WGDs predated evolutionary divergence of the four plant contrasts: maize is a 12 M ya paleo-tetraploid, cotton is a 1.6 M ya meso-tetraploid, and the ~70 M ya *Rho* WGD predated grass diversification. Paralogs derived from these WGDs notoriously obscure contrasts between orthologs in these systems. As such, we treated all genomes as haploid and did not include an outgroup; therefore, orthogroups should not include

**Table 1.** Comparison of synteny and orthogroup methods.

To test the precision of GENESPACE syntenic orthogroups estimates, we contrasted seven pairs of haploid genome assemblies. We present the percent of genes that were found in an orthogroup that hit a single chromosome per genome from the default OrthoFinder and GENESPACE runs. The precision of syntenic block breakpoint estimates was calculated similarly, where the percentage of genes that are placed in a single syntenic block per genome are presented for MCScanX run on all hits, those where both the query and target genes are in the same orthogroup ('OG') or via the GENESPACE pipeline.

| | | (a) % genes in single-copy OGs | | (b) % genes in single-copy syntenic blocks | | |
|---|---|---|---|---|---|---|
| | Age (~M ya) | OrthoFinder | GENESPACE | MCScanX | MCScanX OG | GENESPACE |
| B73 vs. B97 maize* | <0.01 | 51.5 | 73.6 | 50.8 | 79.0 | 93.4 |
| Human Hg38 vs. T2T | 0–0.1 | 87.7 | 95.9 | 81.1 | 95.0 | 97.7 |
| Cotton*,+ | 0.5 | 35.6 | 85.7 | 2.7 | 14.1 | 96.2 |
| HAL2 vs. FIL2 panicgrass* | 1.1 | 74.8 | 83.2 | 62.3 | 89.3 | 92.0 |
| Human-chimpanzee | 7 | 81.1 | 90.2 | 78.6 | 91.2 | 93.3 |
| *Sorghum-Brachypodium** | 50 | 46.7 | 50.2 | 49.3 | 67.4 | 76.3 |
| Human-chicken | 310 | 66.7 | 68.5 | 66.4 | 71.2 | 73.0 |

*The plant genomes all have one or more WGDs that predate divergence of the genomes,.+Cotton species *Gossypium barbadense* and *G. darwinii* have the most recent WGD of ~1.6 M ya, which causes a large number of blocks to be included as two copies; to avoid confusion between subgenomes, blkSize, and nGaps parameters were increased from 5 (default) to 10 genes.

homeologs since the genomes share a common ancestor that arose after the WGD (see *'additional considerations' methods section: Outgroups and the phylogenetic context of orthology inference*). In each run, we contrasted GENESPACE-derived orthogroups and syntenic blocks to the defaults from OrthoFinder and MCScanX, respectively (*Table 1*). Since it is a common practice to refine hits prior to running MCScanX, we also included syntenic block coverage from MCScanX run on orthogroup-constrained hits where both the target and query genes must be in the same orthogroup. To contrast

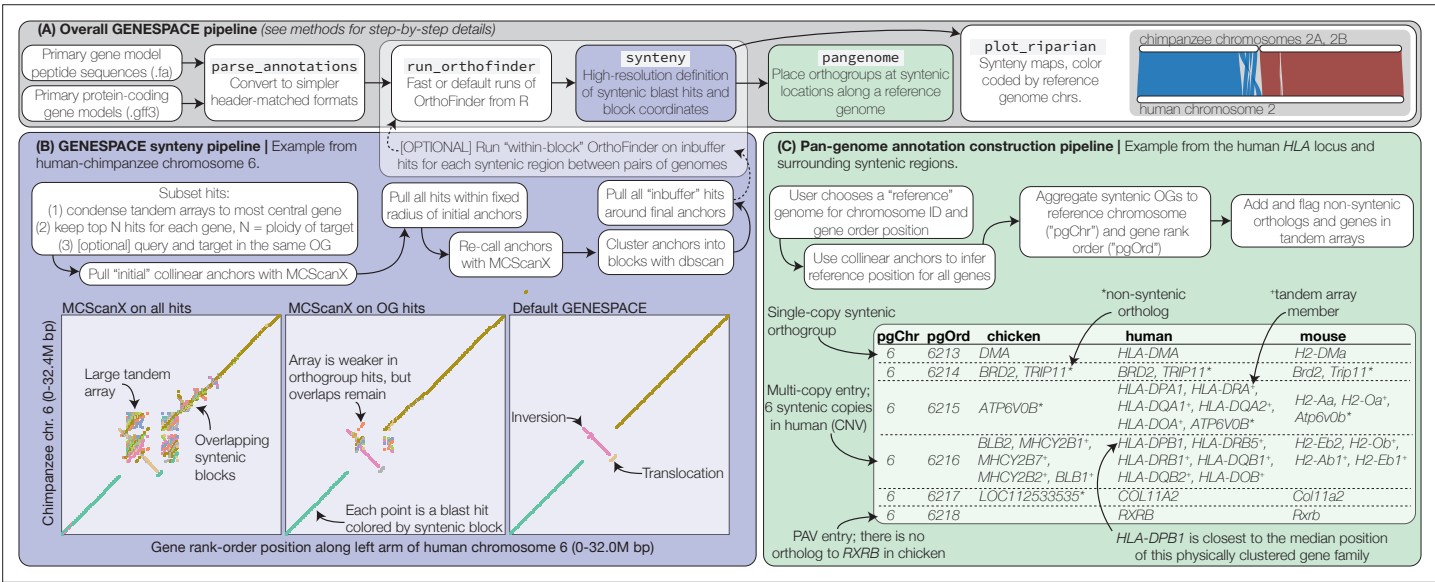

**Figure 1.** GENESPACE synteny and pan-genome annotation methods. (**A**, grey panel) GENESPACE runs and parses OrthoFinder results into a synteny-constrained pan-genome annotation. (**B**, purple panel) Chromosome, gene rank order, and orthogroup membership are added to BLAST hits, which allows direct integration between estimates of orthology and synteny. The three dotplots present the efficacy of GENESPACE syntenic blocks by exploring a particularly challenging region on human (*x*-axis) and chimpanzee (*y*-axis) chr. 6. Each point is a BLAST hit rank-order position, colored by syntenic block; colors are recycled if there are more than eight blocks. (**C**, green panel) Synteny-constrained orthogroups and optionally non-syntenic orthologs are decomposed into a pan-genome annotation where each orthogroup is placed at its inferred syntenic position.

each approach, we calculated the percent of genes in (1) orthogroups and (2) syntenic blocks that were placed on exactly one chromosome per genome (**Table 1**).

For every pair of genomes, GENESPACE produced a greater percentage of single-copy syntenic blocks and genes in single chromosome orthogroups than either OrthoFinder or MCScanX in isolation. GENESPACE also outperformed simple integration between the two methods through MCScanX on orthogroup-constrained hits. The improved performance of GENESPACE synteny-constrained orthogroups was most subtle between highly diverged haploid animal assemblies. For example, in the comparison between human and chicken, GENESPACE resolved 2% and 1.8% more single-copy orthogroups and syntenic blocks than default OrthoFinder and orthogroup-constrained MCScanX, respectively. In contrast, the benefits of GENESPACE were most pronounced in recently diverged genomes with a history of WGDs. Single-copy orthogroups between two meso-tetraploid cotton species that share a WGD that predated speciation by ~1 M ya, were uncommon in the default OrthoFinder run (35.6%) but far more prevalent in GENESPACE syntenic orthogroups (85.7%). Similarly, homeologs derived from the cotton WGD impacted estimates of syntenic blocks: only 14% of the genomes were single-copy syntenic in orthogroup-constrained MCScanX but 96.2% were single copy in GENESPACE syntenic blocks. Combined, these results demonstrate significant flexibility and utility of GENESPACE across a range of evolutionary histories and divergence.

It is important to note that some evolutionary processes, including small-scale translocations, can cause true orthologs to exist outside of syntenic regions. Closely related genomes without a history of WGDs tend to have few non-syntenic orthologs. For example, there are 1096 non-syntenic orthologs (6.3% of all orthologs) between human and chimpanzee. In contrast, the 50 M ya diverged Sorghum and Brachypodium genomes have 9002 non-syntenic orthologs, many of which are the result of over-retained *Rho* WGD-derived paralogs (see below). Since the non-syntenic orthologs can be important in some situations, GENESPACE embraces this complexity by including and flagging non-syntenic orthologs within the pan-genome annotation (**Figure 1**).

## Synteny-anchored exploration of vertebrate sex chromosomes

GENESPACE facilitates the exploration and analysis of sequence evolution across multiple genomes within regions of interest (e.g., quantitative trait loci [QTL] intervals, see the next section). One particularly instructive example comes from the origin and evolution of the mammalian XY and avian ZW sex chromosome systems. To explore these chromosomes, we ran GENESPACE on 15 haploid avian and mammalian genome assemblies, spanning most major clades of birds, placental mammals, monotremes, and marsupials with available chromosome-scale annotated reference genomes (See Materials and methods). We also included two reptile genomes as outgroups to the avian genomes. The heteromorphic chromosomes (Y and W) are often unassembled, or, where assemblies exist, lack sufficient synteny to provide a useful metric for comparative genomics. As such, we chose to focus on the homomorphic X and Z chromosomes, which have remained surprisingly intact over the >100 M years of independent mammalian (**Murphy et al., 1999**) and avian evolution (**Zhou et al., 2014**; **Figure 2**, **Figure 2—figure supplement 1**).

While the same or similar genomic regions often recurrently evolve into sex chromosomes, perhaps due to ancestral gene functions involved in gonadogenesis, evidence about the nonrandomness of sex chromosome evolution is still contentious (**Kratochvíl et al., 2021**). Given our analysis, the avian Z chromosome clearly did not evolve from either of the two reptile Z chromosomes sampled here, but instead likely arose from autosomal regions or unsampled ancestral sex chromosomes. The situation in mammals is less clear, in part because both reptile genomes are more closely related to avian than mammalian genomes, which makes ancestral state reconstructions between the two groups less accurate. Nonetheless, the mammalian X and sand lizard Z chromosomes partially share syntenic orthology, an outcome that would be consistent with common descent from a shared ancestral sex chromosome or autosome containing sex-related genes. The shared 91.7 M bp region between the human X and sand lizard Z represents 59.0% of the human X chromosome genic sequence. The remaining 64.0 M bp of human X linked sequences are syntenic with autosomes in the sand lizard and garter snake genomes (**Figure 2**).

The eutherian mammalian X chromosome is largely composed of two regions, an X-conserved ancestral sex chromosome region that arose in the common ancestor of therian mammals, and an X-added region that arose in the common ancestor of eutherians (**Ross et al., 2005**). Consistent with

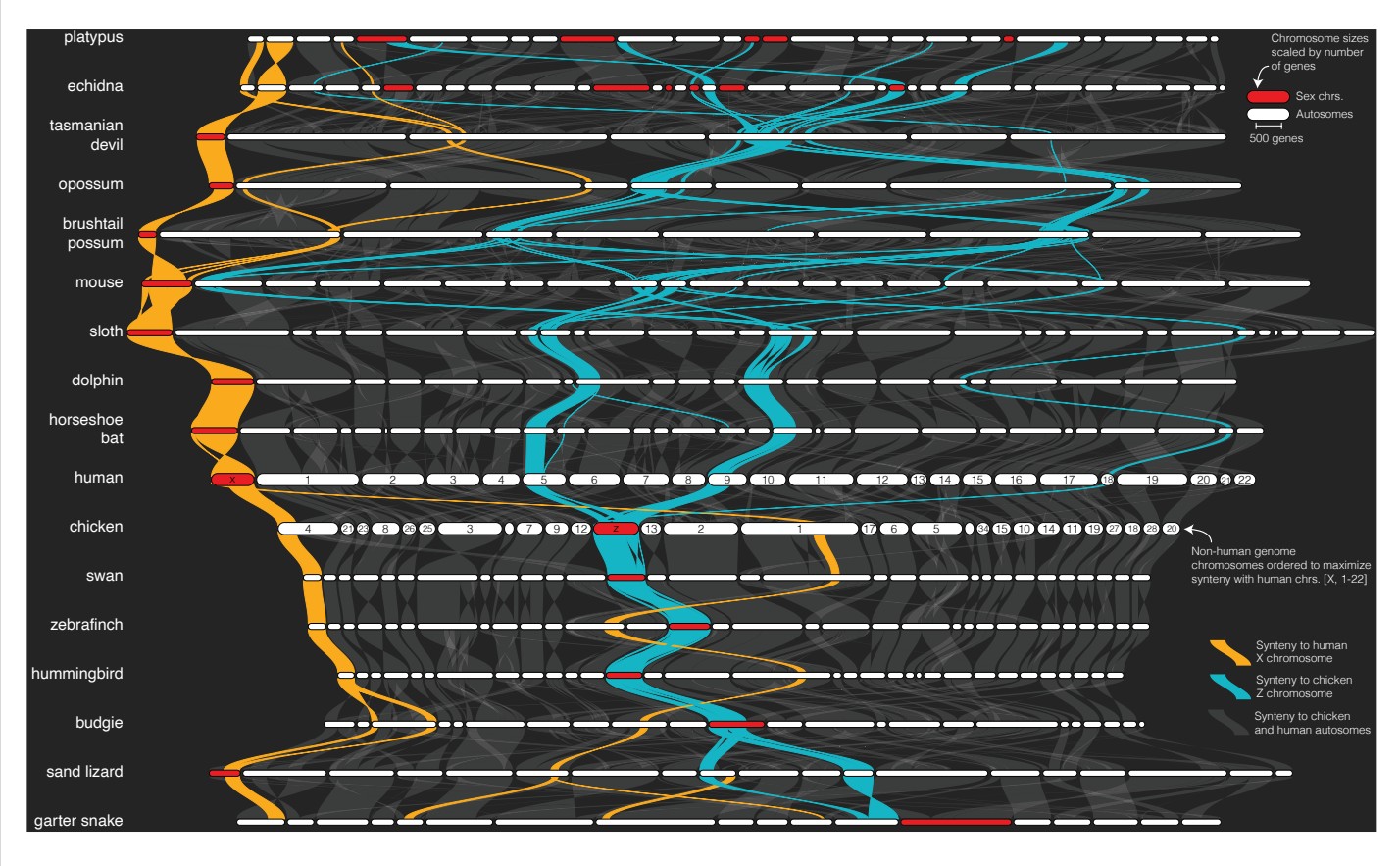

**Figure 2.** Sex chromosome syntenic network across 17 representative vertebrate genomes. The plot was generated by the plot_riparian GENESPACE function. Genomes are ordered vertically to minimize the number of translocations between each pairwise combination. Chromosomes are ordered horizontally to maximize synteny with the human chromosomes [X, 1–22]. Regions containing syntenic orthogroup members to the mammalian X (gold) or avian Z (blue) chromosomes are highlighted. All sex chromosomes are represented by red segments while autosomes are white. Chromosome segment sizes are scaled by the total number of genes in syntenic networks and positions of the braids are the gene order along the chromosome sequence. See *Figure 2—figure supplement 1* for the full synteny graph including autosomes and chromosome labels.

The online version of this article includes the following figure supplement(s) for figure 2:

**Figure supplement 1.** Complete map of synteny, color-coded by synteny with human chromosomes X, 1-22.

this evolutionary history, the X chromosome is syntenic across all five eutherian mammals studied here. Further, a 107.2 M bp (68.8%) segment of the human X, which corresponds with the X-conserved region, is syntenic with 77.8 M bp (93.9%) of the Tasmanian devil X chromosome and represents the entire syntenic region between the human and all three marsupial X chromosomes (*Figure 2*).

Similarly, the chicken Z chromosome is retained in its entirety across all five avian genomes. The only notable exception being the budgie Z chromosome, which features a partial fusion between the Z and an otherwise autosomal 19.5 M bp segment of chicken chromosome 11 (*Figure 2*), potentially representing a neo-sex chromosome fusion that has not yet been described.

In contrast to conserved eutherian and avian sex chromosomes, the complex monotreme XnYn sex chromosomes are only partially syntenic between the two sampled genomes. Only the first X chromosomes are ancestral to both echidna and platypus (*Rens et al., 2007*), and all are unrelated to the mammalian X chromosomes (*Figure 2—figure supplement 1*), consistent with their independent evolution (*Rens et al., 2007*). Interestingly, the entirety of the echidna X4 and 47.6 M bp (67.9%) of the genic region of the platypus X5 chromosomes are syntenic with the avian Z chromosome (*Figure 2*). The phylogenetic scale of the genomes presented here precludes evolutionary inference about the origin of these shared sex chromosome sequences; however, the possibility of parallel evolution of sex chromosomes between such diverged lineages may prove an interesting future line of inquiry.

## Exploiting synteny to track candidate genes in grasses

The Poaceae grass plant family is one of the best studied lineages of all multicellular eukaryotes and includes experimental model species (*Brachypodium distachyon*; *Panicum hallii*; *Setaria viridis*) and many of the most productive (*Zea mays* – maize/corn; *Triticum aestivum* – wheat, *Oryza sativa* – rice) and emerging (*Sorghum bicolor* – sorghum; *Panicum virgatum* – switchgrass) agricultural crops. Despite the tremendous genetic resources of these and other grasses, genomic comparisons among grasses are difficult, in part because of an ancient polyploid origin (see the next section), and because subsequent WGDs are a feature of most clades of grasses. For example, maize is an 11.4 M ya paleo-polyploid (*Gaut and Doebley, 1997*), allo-tetraploid switchgrass formed 4–6 M ya (*Lovell et al., 2021b*), and allo-hexaploid bread wheat arose about 8 k ya (*Haas et al., 2019*). In some cases, homeologous gene duplications from polyploidy have generated genetic diversity that can be targeted for crop improvement; however, in other cases the genetic basis of trait variation may be restricted to sequences that arose in a single subgenome. Thus, it is crucial to contextualize comparative–quantitative genomics and explicitly explore only the orthologous or homeologous regions of interest when searching for markers or candidate genes underlying heritable trait variation — a significant challenge in the complex and polyploid grass genomes.

To help overcome this challenge and provide tools for grass comparative genomics, we conducted a GENESPACE run and built an interactive viewer hosted on Phytozome (*Goodstein et al., 2012*). Owing to its use of within-block orthology and synteny constraints (*Figure 1*), GENESPACE is ideally suited to conduct comparisons across species with diverse polyploidy events. Default parameters produced a largely contiguous map of synteny even across notoriously difficult comparisons like the paleo-homeologs between the maize subgenomes (*Figure 3A*). Furthermore, the sensitive synteny construction pipeline implemented by GENESPACE effectively masks additional paralogous regions like those from the *Rho* duplication that gave rise to all extant grasses.

Breeders and molecular biologists can take two general approaches to understand the genetic basis of complex traits: studying variation caused by a priori-defined genes of interest or determining candidate genes from genomic regions of interest. As an example of the exploration of lists of a priori-defined candidate genes, we analyzed PAV of 86 genes shown to be involved in the transition between $C_3$ and $C_4$ photosynthesis (*Ding et al., 2015*), the latter permitting ecological dominance in arid climates and agricultural productivity under forecasted increased heat load of the next century. To conduct this analysis, we built pan-genome annotations across the seven grasses anchored to $C_4$ maize, which was the genome in which these genes were discovered. This resulted in 159 pan-genome entries: nearly always two placements for each gene in the paleo-tetraploid maize genome. Given that many of these genes were discovered in part because of sequence similarity to genes in *Arabidopsis* and other diverged plant species, it is not surprising that PAV among $C_3/C_4$ genes was lower than the background (9.7% vs. 38.2%, odds ratio = 5.7, $p < 1 \times 10^{-16}$; *Figure 3B*). However, these ratios were highly variable among genomes, particularly among the $C_3$ species (wheat, rice, *B. distachyon*), which had far more absences than the $C_4$ species (15.3% vs. 5.5%, odds ratio = 3.1, $p = 6.25 \times 10^{-8}$, *Figure 3B*). This effect is undoubtedly due in part to the increased evolutionary distance between maize and the $C_3$ species compared to the other $C_4$ species. However, when controlling for the elevated level of absent genes globally in $C_3$ species, the effect was still very strong: the odds of $C_3$ species having more of these $C_3/C_4$ genes at syntenic pan-genome positions than the background was always lower than the $C_4$ species (*Figure 3C*). Despite these interesting patterns, given only a single $C_3/C_4$ phylogenetic split in this dataset, it is not possible to test evolutionary hypotheses regarding the causes of such PAV. Nonetheless, this result suggests a possible role of gene loss or gain as an evolutionary mechanism for drought- and heat-adapted photosynthesis.

Like the exploration of a priori-defined sets of genes, finding candidate genes within QTL intervals usually involves querying a single reference genome and extracting genes with promising annotations or putatively functional polymorphism. In the case of a biparental mapping population genotyped against a single reference, this is a trivial process where genes within physical bounds of a QTL are the candidates. However, many genetic mapping populations now have reference genome sequences for all parents; this offers an opportunity to explore variation among functional alleles and PAV, which would be impossible with a single reference genome. GENESPACE is ideally suited for this type of exploration, and indeed was originally designed to solve this problem between the two *P. hallii* reference genomes and their $F_2$ progeny (*Lovell et al., 2018*) using synteny to project the positions of

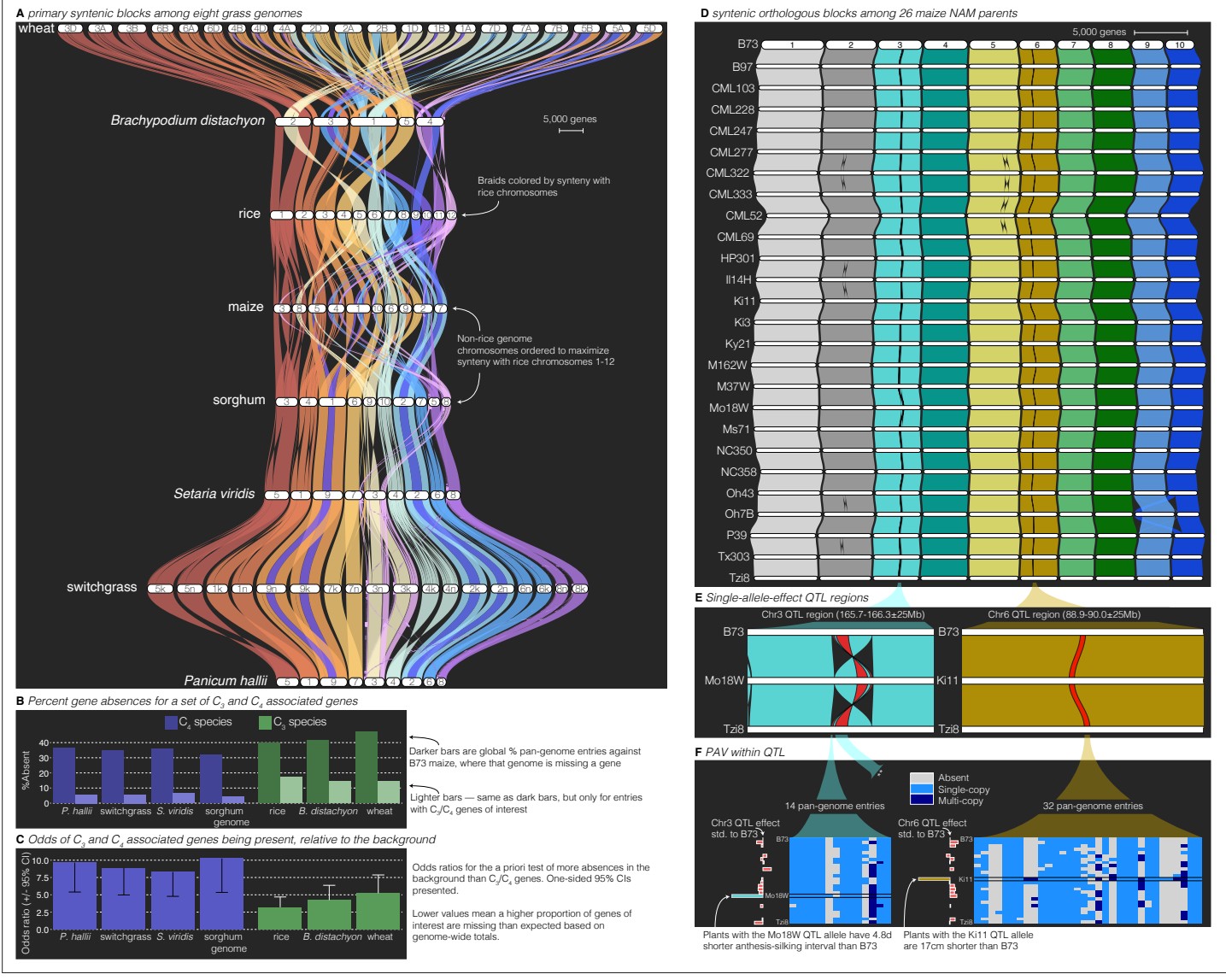

**Figure 3.** Comparative–quantitative genomics in the grasses. (**A**) The GENESPACE syntenic map ('riparian plot') of orthologous regions among eight grass genomes. Chromosomes are ordered horizontally to maximize synteny with rice and ribbons are color coded by synteny to rice chromosomes. Genomes are ordered vertically by general phylogenetic positions. (**B**) The upper bars display the proportion of maize gene models without syntenic orthologs ('absent') in each genome, split by the full background (dark colors) and 86 $C_3$/$C_4$ genes (light colors). (**C**) The proportion of absent genes is higher in the $C_3$ genomes (green bars), even when controlling for more global gene absences (lower odds ratios). (**D**) Syntenic orthologs, excluding homeologs among the 26 maize nested association mapping (NAM) founder genomes, with two quantitative trait loci (QTL) intervals highlighted on chromosome 3 ('Chr3') and chromosome 6 ('Chr6'). (**E**) Focal QTL regions that affect productivity in drought where only the genome that drives the QTL effect (middle), the top (B73) and bottom (Tzi8) genomes are presented and the region plotted is restricted to the physical B73 QTL interval and a 25 M bp buffer on either side. Note that the Chr3 QTL disarticulates into two intervals. Due to a larger number of potential candidate genes, the larger Chr3 region, flagged with **, is explored separately in *Figure 3—figure supplement 1*. (**F**) Presence–absence and copy number variation are presented for two of the three intervals as heatmaps where each row is a genome (order following panel D), each column is a pan-genome entry (see *Figure 1*), and the color of each tile indicates absence (gray), single copy (light blue), and multicopy (dark blue). PAV/CNV of the focal genome is outlined. For each interval, the estimated QTL allelic effect relative to B73 of each genome is plotted as bars to the right of the heatmap.

The online version of this article includes the following figure supplement(s) for figure 3:

**Figure supplement 1.** Map of PAV in the larger MO18W chromosome 3 QTL.

genes across multiple genomes onto the physical positions of a reference. To illustrate this approach, we reanalyzed QTL generated from the 26-parent USA maize nested association mapping (NAM) population (*Li et al., 2016*). Originally, candidates for these QTL were defined by the proximate gene models only in the B73 reference genome (*Li et al., 2016*); however, with GENESPACE and the recently released NAM parent genomes (*Hufford et al., 2021*), it is now possible to evaluate candidate genes present in the genomes of other NAM founder lines but either absent or unannotated in the B73 reference genome.

To explore this possibility, we built a single-copy synteny map across all 26 NAM founders anchored to the B73 genome (*Figure 3D*). We opted to focus on QTL where the allelic effect of a single parental genome was an outlier relative to all other alleles. Such 'private' allelic contributions in multi-parent populations offer a powerful opportunity to define high-confidence candidates as genes with parent-specific sequences that match parent-specific allelic contributions to phenotypic trait variation (*Abdulkina et al., 2019*). Among the 190 QTLs, three displayed outlier effects of one parent: cultivar 'Mo18W' contributed an allele for delayed anthesis-silking interval at two adjacent chromosome 3 ('Chr3') QTLs and cultivar 'Ki11' contributed an allele that reduced plant height under drought at a QTL on chromosome 6 ('Chr6') (*Li et al., 2016*). Given that these QTL were chosen only due to their parental allelic effects, we were surprised to find that the two Mo18W QTL regions exist within a 11.7 M bp derived inversion that is only found in the Mo18W genome (*Figure 3D, E*). Since inversions reduce recombination, it is possible that multiple Mo18W causal variants have been fixed in linkage disequilibrium in this NAM population.

In addition to this chromosomal mutation and sequence variation between the parents and B73 (*Li et al., 2016*), we sought to define candidate genes from the patterns of presence–absence and copy number variation, explicitly looking for genes that were private to the focal genome. Two genes in the smaller Chr3 and one gene in the larger Chr3 interval were private to Mo18W, and four genes in three pan-genome entries (one two-member array) were private to Ki11 in the Chr6 interval (*Figure 3F*, *Figure 3—figure supplement 1*). While these genes do not have functional annotations relating to drought, this method provides additional candidates that would not have been discovered by B73-only candidate gene exploration.

## Studying the WGD that led to the diversification of the grasses

Like most plant families (*Barker et al., 2016*; *Stebbins, 1950*; *One Thousand Plant Transcriptomes Initiative, 2019*), but unlike nearly all animal lineages (*Muller, 1925*), the grasses radiated following a

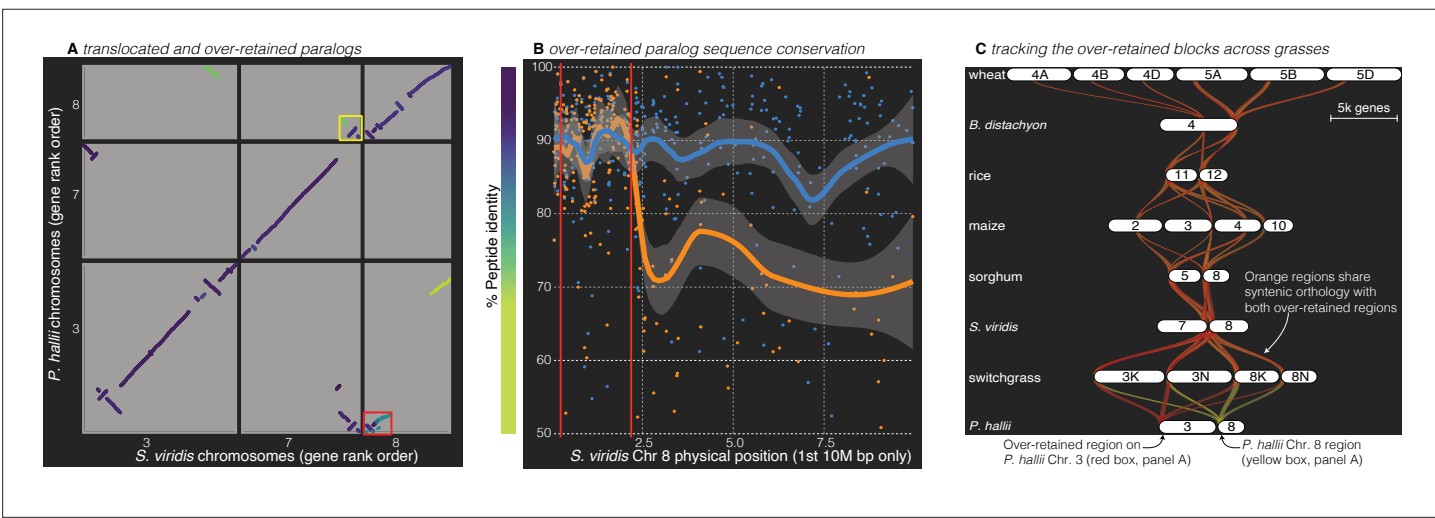

**Figure 4.** Analysis of the grass *Rho* WGD. (**A**) BLAST hits between *P. hallii* and *S. viridis* where the target and query genes were in the same orthogroup are plotted and color coded by sequence similarity. Two over-retained regions are highlighted in the red and yellow boxes. (**B**) The protein identity of *S. viridis* chromosome 8 primary orthologous (blue line) hits against *P. hallii* chromosome 8 and the secondary hits (orange line) against *P. hallii* chromosome 3 demonstrate sequence conservation heterogeneity. The region between the two red vertical lines corresponds to the red-boxed over-retained primary block in panel A. (**C**) The two boxed regions in panel A were tracked from *P. hallii* chromosomes 3 (red) and 8 (yellow); 50% transparency of the braids means that overlapping regions appear orange.

whole-genome duplication: the ~70 M ya *Rho* WGD. The resulting gene family redundancy and gene-function subfunctionalization is hypothesized to underlie the tremendous ecological and morphological diversity of grasses (*Preston et al., 2009*; *Preston and Kellogg, 2006*; *Wu et al., 2008*).

To explore sequence variation among *Rho*-derived paralogs, we used GENESPACE to build a ploidy-aware syntenic pan-genome annotation among eight species, using the built-in functionality that allows the user to mask primary (likely orthologous) syntenic regions and search for secondary hits (likely paralogous, *Figure 4A*). This method acts similarly to using an outgroup that predated the WGD (see *'additional considerations' methods*), but using the same OrthoFinder run as in *Figure 3A*. Overall, the peptide identity between *Rho*-derived paralogous regions was much lower than orthologs between species (e.g., *S. viridis* vs. *P. hallii*: Wilcoxon $W = 88{,}094{,}632$, $p < 1 \times 10^{-16}$), consistent with the previous discovery that the *Rho* duplication predated the split among most extant grasses (*Ma et al., 2021*). However, as has been previously observed (*Wang et al., 2011*), there is significant variation in the relative similarity of *Rho*-duplicated sequences. As an example, the peptide sequences of single-copy gene hits in primary syntenic regions (median identity = 90.6%) between chromosome 8 of *P. hallii* and *S. viridis*, were 26.9% more similar than the secondary *Rho*-derived regions (*Figure 4B*; median identity = 71.4%, Wilcoxon $W = 87{,}842$, $p < 1 \times 10^{-16}$). However, *S. viridis* chromosome 8 contained a single over-retained paralogous region. Unlike all other *Rho*-derived blocks, the *P. hallii* paralogs to this 2.7 M bp chromosome 8 region were not significantly less conserved than the primary orthologous region (91.6% vs. 91.9%, $W = 14{,}830$, $p = 0.13$). Outside of this region, the peptide identity of paralogs returned to the genome-wide average (*Figure 4B*). In line with this observation, the GENESPACE run treating the eight genomes as haploid representations could not distinguish between the *Rho*-derived paralogs in the over-retained region across all grasses (*Figures 3A and 4C*), except for of all chromosome pairs between *B. distachyon*, wheat, and blocks connecting Maize chromosome 10 to sorghum chromosome 5.

It is interesting to note that all syntenic over-retained regions were at the extreme termini of the chromosomes outside of maize, *B. distachyon* and wheat; further, the only genome with complete segregation of the two paralogs, wheat, also retains these regions in the center of all six chromosomes (*Figure 4C*). These results are consistent with the proposed evolutionary mechanism (*Wang et al., 2011*) where concerted evolution and 'illegitimate' homeologous recombination may have homogenized these paralogous regions. This process would be less effective in pericentromeric regions than the chromosome tails, where a single crossover event would be sufficient to homogenize two paralogous regions.

## Conclusions

Combined, the historical abundance of genetic mapping studies and ongoing proliferation of genome resources provides a strong foundation for the integration of comparative and quantitative genomics to accelerate discoveries in evolutionary biology, medicine, and agriculture. The incorporation of synteny and orthology into comparative genomics and quantitative genetics pipelines offers a mechanism to bridge these disparate disciplines. Here, we presented the GENESPACE R package as a framework to help bridge the current gaps between comparative and quantitative genomics, especially in complex evolutionary systems. We hope that the examples presented here will inspire further work to leverage the powerful genome-wide annotations that are coming online, both within and among species.

## Materials and methods
### GENESPACE pipeline and analysis overview

GENESPACE operates on gff3-formatted annotation files and accompanying peptide fasta files for primary gene models. There are convenience functions for reformatting the gff and peptide files to simplify the naming scheme and reduce redundant gene models to the primary longest transcript. With these data in hand, GENESPACE calculates BLAST-like hits from DIAMOND2 and runs OrthoFinder (*Emms and Kelly, 2019*) to infer orthogroups and orthologues. GENESPACE then extracts syntenic regions from the hits using a combination of graph- and cluster-based approaches, producing syntenic orthogroups for each unique (not reciprocal) pair of genomes. Syntenic orthogroups are then collapsed into a pan-genome annotation, which is a matrix of positions against a reference genome (rows) and

unique gene models in each syntenic orthogroup for each genome (columns). Detailed step-by-step pipeline methods can be found below.

All analyses were performed in R 4.1.2 on macOS Big Sur 10.16. The following R packages were used for visualization or within GENESPACE v0.9.3 (February 11, 2022 release): data.table v1.14.0 (*Dowle and Srinivasan, 2021*), dbscan v1.1-8 (*Hahsler et al., 2019*), igraph v1.2.6 (*Csardi and Nepusz, 2006*), Biostrings v2.58.0 (*Pagès et al., 2020*), and rtracklayer v1.50.0 (*Lawrence et al., 2009*). GENESPACE also calls the following third party software: DIAMOND v2.0.8.146 (*Buchfink et al., 2021*), OrthoFinder v2.5.4 (*Emms and Kelly, 2019*), and MCScanX no version installed on October 23, 2021 (*Wang et al., 2012*). All results were generated programmatically; the accompanying scripts and key output are available on github: jtlovell/GENESPACE_data. Minor adjustments to figures to improve clarity were accomplished in Adobe Illustrator v26.01. A full description of each step in GENESPACE is provided in the documentation that accompanies the package source code on github (jtlovell/GENESPACE).

## Description of the vignettes

Publicly available genome annotations were downloaded on or before October 8, 2021. See *Table 2* for data sources, citations, and metadata. All GENESPACE runs used default parameterization, with the following exceptions: (1) the *Rho* grass run allowed a single secondary hit (default is 0, this is how the paralogs are explicitly searched for) and maximum number of gaps in secondary regions of 10 (default is 5, relaxed to reduce ancient paralogous block splitting), and (2) the maize run used the 'fast' OrthoFinder method since all genomes are closely related and haploid. Some maize genomes contained small alternative haplotype scaffolds, which were dropped for all analyses.

The publicly available $C_3$/$C_4$ gene lists and QTL intervals were generated against the v2 maize assembly. To make this comparable to the across grass and NAM parent GENESPACE runs, we also accomplished a fast GENESPACE run between v2 and the two v5 versions used here. The orthologs and syntenic mapping between these versions are included as text files in the data repository.

All statistics presented here were calculated within R. To compare nonnormal distributions (e.g., sequence identity), we used the nonparametric signed Wilcoxon ranked sum test. To measure sequence divergence, we calculated ungapped percent peptide sequence identity from pairwise Needleman–Wunsch global alignments, implemented in Biostrings (*Pagès et al., 2020*). To determine single outliers from a unimodal distribution, we applied the Grubbs test implemented in the outliers R package (*Komsta, 2011*). Some figures were constructed outside of GENESPACE using base R plotting routines and ggplot2 v3.3.3 (*Wickham, 2016*). Some color palettes were chosen with RColorBrewer (*Neuwirth, 2014*) and viridis (*Garnier et al., 2021*).

## GENESPACE pipeline: estimating syntenic orthogroups

GENESPACE relies on high-confidence homologs, which are typically defined as members of the same orthogroup via OrthoFinder. In addition to the original method, which clusters genes and builds a graph from closely related genes based on BLAST scores (*Emms and Kelly, 2015*), OrthoFinder can also use gene trees to split orthogroups into pairwise orthologs (*Emms and Kelly, 2019*), which represent a stricter definition of orthology. GENESPACE attempts to merge the benefits of each of these methods by first only considering orthogroups for synteny, which allows users to optionally include paralogs in the scan, then including non-syntenic gene tree-inferred orthologs into the pan-genome annotation during its final steps (*Figure 1*, see pan-genome pipeline description below). The initial orthogroups are inferred 'globally' using pairwise reciprocal hits, either using the default OrthoFinder specification or 'fast' by only looking at unique pairs of genomes. Depending on the ploidy and user specifications, orthogroups can be re-estimated within syntenic regions. These three steps are detailed below.

*Method 1: default global orthogroups*. The default behavior of GENESPACE is to run OrthoFinder using its default parameters. GENESPACE then builds a vector of OrthoFinder geneIDs and their corresponding orthogroup membership. When synteny is called, the GENESPACE-formatted gff text files are read in and merged with OrthoFinder sequenceIDs.

*Method 2: fast orthogroup estimation*. GENESPACE offers a 'fast' orthofinderMethod (*Table 3*), which performs only one-way DIAMOND2 searches, where the genome annotation with more gene models serves as the query and the smaller annotation is the target. The hits then are mirrored, and

**Table 2.** Raw data sources.

A list of the genomes used in analyses here. Genome version IDs are taken from those posted on the respective data sources and may not reflect the name of the genome in the publication. Where multiple haplotypes are available, only the primary was used for these analyses. All polyploids presented here have only a primary haplotype assembled into chromosomes.

| ID | Species | Genome version | Data source | Ploidy* | Reference |
|---|---|---|---|---|---|
| garter snake | *Thamnophis elegans* | rThaEle1.pri | NCBI | 1 | *Rhie et al., 2021* |
| sand lizard | *Lacerta_agilis* | rLacAgi1.pri | NCBI | 1 | *Rhie et al., 2021* |
| chicken | *Gallus gallus* | mat.broiler.GRCg7b | NCBI | 1 | https://www.ncbi.nlm.nih.gov/grc |
| hummingbird | *Calypte anna* | bCalAnn1_v1.p | NCBI | 1 | *Rhie et al., 2021* |
| budgie | *Melopsittacus undulatus* | bMelUnd1.mat.Z | NCBI | 1 | Unpublished VGP |
| swan | *Cygnus olor* | bCygOlo1.pri.v2 | NCBI | 1 | *Rhie et al., 2021* |
| zebra finch | *Taeniopygia guttata* | bTaeGut1.4.pri | NCBI | 1 | *Rhie et al., 2021* |
| echidna | *Tachyglossus aculeatus* | mTacAcu1.pri | NCBI | 1 | *Zhou et al., 2021* |
| platypus | *Ornithorhynchus anatinus* | mOrnAna1.pri.v4 | NCBI | 1 | *Zhou et al., 2021* |
| brushtail possum | *Trichosurus vulpecula* | mmTriVul1.pri | NCBI | 1 | *Rhie et al., 2021* |
| opossum | *Monodelphis domestica* | MonDom5 | NCBI | 1 | *Mikkelsen et al., 2007* |
| Tasmanian devil | *Sarcophilus harrisii* | mSarHar1.11 | NCBI | 1 | *Rhie et al., 2021* |
| human (Hg38) | *Homo sapiens* | GRCh38.p13 | NCBI | 1 | https://www.ncbi.nlm.nih.gov/grc |
| human (t2t) | *Homo sapiens* | CHM13-T2T v2.1 | NCBI | 1 | *Nurk et al., 2022* |
| chimpanzee | *Pan troglodytes* | Clint_PTRv2 | NCBI | 1 | *Chimpanzee Sequencing and Analysis Consortium, 2005* |
| mouse | *Mus musculus* | GRCm39 | NCBI | 1 | https://www.ncbi.nlm.nih.gov/grc |
| dog | *Canis lupus familiaris* | Dog10K_Boxer_Tasha | NCBI | 1 | *Jagannathan et al., 2021* |
| sloth | *Choloepus didactylus* | mChoDid1.pri | NCBI | 1 | *Rhie et al., 2021* |
| horseshoe bat | *Rhinolophus ferrumequinum* | mRhiFer1_v1.p | NCBI | 1 | *Rhie et al., 2021* |
| dolphin | *Tursiops truncatus* | mTurTru1.mat.Y | NCBI | 1 | Unpublished VGP |
| *P. hallii* | *Panicum hallii var. hallii* | HAL2_v2.1 | Phytozome | 1 | *Lovell et al., 2018* |
| *P. hallii (FIL)* | *Panicum hallii var. filipes* | FIL2_v3.1 | Phytozome | 1 | *Lovell et al., 2018* |
| switchgrass | *Panicum virgatum* | AP13_v5.1 | Phytozome | 2 | *Lovell et al., 2021b* |
| *S viridis* | *Setaria viridis* | v2.1 | Phytozome | 1 | *Mamidi et al., 2020* |
| *Sorghum* | *Sorghum bicolor* | BTx623_v3.1 | Phytozome | 1 | *Paterson et al., 2009* |
| maize | *Zea mays* | B73_refgen_v5 | NCBI | *2 | *Hufford et al., 2021* |
| rice | *Oryza sativa cv 'kitaake'* | kitaake_v2.1 | Phytozome | 1 | *Jain et al., 2019* |
| *Brachypodium* | *Brachypodium distachyon* | Bd21_v3.1 | Phytozome | 1 | *International Brachypodium Initiative, 2010* |
| wheat | *Triticum aestivum* | V4 (Chinese Spring) | NCBI | 3 | *Zhu et al., 2021* |
| *G barbadense* | *Gossypium barbadense* | v1.1 | Phytozome | 2 | *Chen et al., 2020* |
| *G. darwinii* | *Gossypium darwinii* | v1.1 | Phytozome | 2 | *Chen et al., 2020* |
| 26 NAM parents | *Zea mays* | see data on NCBI | NCBI | *1 | *Hufford et al., 2021* |

*Ploidy indicates how the genome was treated in the analyses. All values match the ploidy of the primary assembly haplotype except maize, where the refgen_v5 was treated as diploid (to match both homeologs) in the multispecies run, but as haploid in the nested association mapping (NAM) founder population to track only meiotic homologs across the population. This parameterization is to match the phylogenetic position of the whole-genome duplication (WGD) in the terminal branch of the grass-wide analysis, but ancestral in the 26-NAM analysis.

**Table 3.** Comparison of GENESPACE setting performance.

The mirrored 'fast' method significantly speeds up OrthoFinder runs by calling DIAMOND2 on each nonredundant pairwise combination of genomes. However, this approach is less sensitive than the default performance and is suggested for only closely related haploid genomes, as the recall of 2:2:2 OGs is less sensitive than the default specification.

| | Default OrthoFinder | GENESPACE 'fast' |
|---|---|---|
| n.1:1:1 OGs | 22,050 | 22,444 |
| n.2:2:2 OGs | 13,793 | 13,511 |
| n.tandem arrays | 10,597 (4433) | 10,599 (4426) |
| *Run time (min) | 59.95 | 12.45 |

*Run time is for ortholog/orthogroup inference (not the GENESPACE pipeline as a whole) using the three cotton genomes, running on 6 2 Gb cores.

each stored as OrthoFinder-formatted BLAST text files. OrthoFinder is then run to the orthogroup step on the precomputed BLAST files. This method results in significant speed improvements with little loss of fidelity among closely related haploid genomes. The user can also specify the sensitivity of DIAMOND2 via the diamondMode parameter during GENESPACE initialization.

*Method 3: orthogroups within syntenic blocks*. In addition to the above global OrthoFinder runs, GENESPACE can rerun OrthoFinder (only through the -og step) in syntenic regions between pairs of genomes. This is accomplished through four additional steps in the synteny pipeline: (1) syntenic hits are split by large syntenic regions; (2) hits from each syntenic region are passed to OrthoFinder; (3) the within-block orthogroups assignments are aggregated into a single vector across all genomes; and finally, (4) the synteny pipeline (see below) is rerun using the newly defined combined syntenic and inBlkOG vector. While using significant computational resources, this process can improve the sensitivity of orthogroup discovery in paralogous or homeologous regions; however, it is not clear that this offers much improvement over global orthogroups between purely haploid genomes. As such, the default behavior of GENESPACE is to only use orthofinderInBlk when any genome has a ploidy >1.

## GENESPACE pipeline: extracting syntenic blast hits

The GENESPACE function 'synteny' accepts several user-defined parameters, which allow for flexibility; however, the defaults are sufficient for most high-quality genomes and evolutionary scenarios. For example, we used the same default parameters for 300 M years of vertebrate evolution, 50 M years and multiple WGDs of grasses, and 10 k years of Maize divergence. For a full list of parameters, see documentation of the GENESPACE function 'set_syntenyParams', but here, we will discuss (1) the minimum number of unique hits within a syntenic block ('blkSize', default = 5), (2) the maximum number of gaps within a block alignment ('nGaps', default = 5), and (3) the radius around a syntenic anchor for a hit to be considered syntenic ('synBuff', default = 100). Synteny begins by processing all gene annotations (steps 1–3), then proceeds to process blast hits for each unique pair of genomes (steps 4–7).

*Step 1: flag the gff3-formatted annotation*. GENESPACE adds the following information to a data matrix that initially contains the name and physical coordinates of each gene for each genome: (1) OrthoFinder IDs (from the sequenceIDs.txt file), (2) gene rank order, and (3) orthogroup IDs.

*Step 2: quality control the annotation*. Chromosomes with fewer than blkSize genes are dropped so that they will not be used for synteny inference.

*Step 3: find and parse tandem arrays*. Tandem arrays are defined through six steps: (1) define potential tandem arrays as orthogroups containing >1 gene for each genome-by-chromosome combination; (2) calculate the maximum gene rank-order gap between each adjacent gene in a potential tandem array; (3) split those with a gap between genes > synBuff into separate arrays using dbscan; (4) flag clusters of >1 genes within synBuff as tandem arrays; (5) choose representative genes for each tandem array as the gene closest to the median position of the array and secondarily as the longest peptide sequence; and (6) recalculate 'arrayOrd' as the gene rank order of tandem array representative genes.

*Step 4: read and process raw blast files.* To reduce unnecessary computation, GENESPACE concatenates both reciprocal BLAST hit files so that all unique combinations of query and target genes are represented in a single matrix. For simplicity, the genome with more gene models is treated as the query and the genome with fewer, the target. For each line ('hit') in the BLAST file, the following data and statistics are added for both the query (genome1) and target (genome2) gene: (1) positional information, including gene rank order; (2) tandem array representatives; (3) hits where both the query and target are in the same orthogroup; and (4) the relative strength of a hit, where scrRank = 1 represents the single highest bit score blast hit for each query and target gene.

*Step 5: define initial syntenic anchors.* The following additional parameters are used to define initial syntenic anchors: nHits1 (the top *n* hits for each gene in the query genome, default ploidy of target genome), nHits2 (the top *n* hits for each gene in the target genome, default = ploidy of query genome), onlyOgAnchors (logical specifying whether anchors hits must be isOg, default = TRUE), and maskTheseHits (what hits should be masked in the search, default = none; see below on modifications for secondary hits and polyploid self hits). These parameters are applied to find regions that are large and collinear enough to be classified as syntenic through three steps: (1) potential anchor hits are defined as those where both the query and target are array representatives, the hit is not in maskTheseHits, the query scrRank ≤ nHits1, the target scrRank ≤ nHits2, and, optionally if onlyOgAnchors, both the query and target are in the same orthogroup; (2) gene rank order is recalculated for the potential anchor hits; (3) MCScanX_h (-s = blkSize, -m = nGaps) is called from R using condensed gene rank-order positions as the physical location, and those hits within collinear blocks are flagged as initial anchors.

*Step 6: clean up initial syntenic anchors.* In some cases, syntenic anchors from step 5 can be broken up or do not extend to the ends of syntenic regions. To resolve this, we run five additional block finalization steps: (1) potential anchor hits within synBuff of the initial anchors are extracted and collinear hits are recalled into 'cleaned' anchors; (2) the cleaned anchor hits are clustered into blocks via dbscan using synBuff as the radius and blkSize as minPts arguments, dropping blocks with fewer anchors than blkSize and merging nearby syntenic regions; (3) final syntenic anchors are flagged by rerunning MCScanX within each syntenic region; (4) initial block breakpoints are generated by dbscan first with a large radius of synBuff, then on reranked gene order within regions with a radius of blkSize; and (5) overlapping blocks that are not duplicated are split using run-length equivalent decoding. Since the final coding of blocks is conducted within broad syntenic regions, the parameters passed should be robust to variation in ploidy and sequence divergence between genomes.

*Step 7: flag anchors, syntenic region hits, and block coordinates.* With the final syntenic anchors from step 6 in hand we finalize and annotate the tab-delimited blast hits. We define two sets of syntenic qualifications: blocks, which are fine grained runs of completely collinear hits, and 'regions', which are clustered blocks that average across minor inversions. Blocks are defined from the step 6 anchors. Anchor hits are reclustered via dbscan with a radius of synBuff and each cluster between each pair of genomes and chromosomes is assigned a 'regID'. All hits within synBuff of an anchor hit are flagged as 'inBuffer'. The coordinates of blocks and regions are calculated from the bounding anchors for each syntenic block.

*Modifications:* steps 5–7 are modified depending on whether the BLAST hit file is intra- or intergenomic, the ploidy of the query and target genome, and the user-defined number of 'secondary hits'. These modifications are as follows. (1) *step 5–7(1): [if necessary] syntenic regions are called for self blast hits within haploid genomes.* This is the simplest case where the steps are ignored and syntenic anchors are defined as self hits between tandem array representatives. Block and region IDs are chromosome IDs. (2) *Modification step 5–7(2): [if necessary] syntenic regions are called for self blast hits within genomes with ploidy >1.* Here, the modified syntenic hits from 5-7(1) are specified in maskTheseHits with a synBuff radius of 500 genes. This excludes potentially problematic large tandem arrays in some genomes. The number of hits after masking is set to ploidy – 1. Then the standard step 5–7 processes are run. (3) *Modification step 5–7(3): [if necessary] syntenic regions are called for blast hits where nSecondaryHits >0.* Here, the methods for syntenic hit calculation from 5-7(2) (if intragenomic) or 5-7 (as a mask of the rerun if intergenomic) are conducted except that the parameters specified are given as those ending in 'Second'.

## GENESPACE pipeline: constructing pan-genome annotations

The GENESPACE function 'pangenome' decodes pairwise syntenic orthologs into a multigenome pan-annotation. The output is a long-formatted text file, where each gene is given a reference genome syntenic position and chromosome, and flags that are described below. In addition, pangenome also returns a wide-formatted data.table (R object) where each row is a pan-genome entry with positional information and each column is a list of genes by genome (e.g., *Figure 1*).

*Step 1: build a reference-anchored map of syntenic hits*. The chromosome and positions of the tandem array representatives from the reference genome annotation form the foundation of mapping across all genomes. Upon building this positional backbone, syntenic hits are pulled for each genome and placed against the anchor genes that are in the same orthogroup.

*Step 2: interpolate syntenic reference positions*. In most pairwise comparisons between genomes, some syntenic orthogroups will be missing in the reference. Since we want to extract PAV by physical coordinates, all syntenic orthogroups, even those that do not include a reference genome gene model, need to have reference positional information. To fill this gap, GENESPACE interpolates the syntenic reference position of all array representative genes in all genomes through a three step pipeline: (1) subset the syntenic anchor hits to ungapped collinear hits following a 1:1 synteny mapping ratio (i.e., perfect diagonal of hits by gene rank order); (2) cluster the 1:1 syntenic anchors so that any jump of >1 gene is split into its own cluster; (3) fill in missing positions through linear interpolation.

*Step 3: determine the reference mapping positions of each syntenic orthogroup*. In the case where the reference genome is purely haploid with no segmental duplicates, this is a straightforward step: the orthogroups that contain reference genome genes are placed at that backbone position, and those without a reference gene are clustered into the most likely interpolated placement. However, we do not want to rely on single-copy references, so GENESPACE allows multiple placements not only for each nonreference gene, but also for genes in the reference genome itself. For example, in a polyploid, we would want to know if there is PAV between two homeologous chromosomes. To do this, we take a three-step approach: (1) the interpolated (or actual reference-anchored) positions of genes in each syntenic orthogroup are extracted and split by interpolated chromosome and maximum gap between any two inferred positions >synBuff; (2) unsplit orthogroups are set aside; (3) split orthogroups are checked and retained if they contain ≥ propAssignThresh (default = 0.5) proportion of genes in the syntenic orthogroup with an interpolated position in that cluster; (4) clustered positions are dropped if there are more than maxPlacementsPerRefChr (default = 2) positional placements for that syntenic orthogroup, retaining the top clusters ranked by propAssignThresh; (5) the culled positional clusters are merged with those from step (2) to create the initial pan-genome annotation.

*Step 4: add and flag other forms of orthogroups*. The reference pan-genome built in step 3 only acts on direct syntenic blast hits (edges), which allows for strict construction of interpolated positions without the potential polluting positional effects of orthogroups in minor or miss-assembled syntenic blocks. GENESPACE fixes these and other complexities of syntenic orthogroups through a four step pipeline to finalize the pan-genome annotation: (1) 'indirect syntenic' orthogroup members are added back into the initial pan-genome annotation by parsing the gff-like text file that contains a vector of syntenic orthogroups; (2) syntenic orthogroups that are missing all interpolated positions are added and given NA positions; (3) tandem array members that are not representative genes are added into the pan-genome annotation; (4) if available, orthologues from the initial OrthoFinder run that are not present in the pan-genome annotation are added and flagged. Note that if OrthoFinder was run using the 'fast' orthofinderMethod, orthologs will not be produced nor added into the pan-genome annotation.

## Additional considerations for comparative genomics parameterization

There are several factors to consider when constructing your GENESPACE run, generating syntenic block breakpoints, or looking at comparative genomics through protein similarity estimates like OrthoFinder. We detail two of these below:

*(1) Outgroups and the phylogenetic context of orthology inference*. OrthoFinder defines orthogroups as the set of genes that are descended from a single gene in the last common ancestor of all the species being considered. As such, the scale of the run matters, often significantly. For example, an orthogroup would not be likely to contain homeologs across the two ancient

subgenomes for a run that included only two maize genomes. Since the coalescence of any two maize genotypes occurred more recently than the ~12M ya WGD, few homeologs would both be descended from the same common ancestor when considering only maize genotypes. Hence, the within-maize NAM parent run (*Figure 3D*) excludes homeologs. However, if an outgroup to maize was included in the run, both maize homeologs would be likely to show common ancestry to a single gene in the outgroup, thus connecting the maize homeologs into a single orthogroup. Hence, both maize homeologous regions are present in the across grasses synteny graph (*Figure 3A*) despite using identical synteny parameters to the maize NAM parent run. Given the potentially significant role of outgroups on the results of the global run, GENESPACE offers an 'outgroup' parameter, which specifies the genomes that should be included in the orthofinder run but excluded from all downstream analyses.

(2) *Studying homoelogs in polyploids and other paralogs.* Non-homeologous paralogs are typically excluded in the default GENESPACE parameterization. This is in part because GENESPACE was originally designed for work with plants, and most plant lineages have undergone one or more WGDs. However, there are two ways to study paralogs by altering GENESPACE parameters: (1) specify an 'outgroup' genome (see above), which is only used for the global OrthoFinder run and set the genomes' ploidy as that expected by the number of duplications relative to the outgroup; (2) if an outgroup is not available (or too distantly related to be of use), specify the synteny parameter 'nSecondaryHits' as the number of paralogous copies per genome. In the second case, secondaryHits are inferred after masking out the normal syntenic hits, then synteny is rerun without requiring anchors to be global orthogroups. In both cases, it will be far better to set orthofinderInBlk to TRUE, so that pairs of genomes are considered and genes without a solid hit in the outgroup are not excluded.

## Acknowledgements

The GENESPACE pipeline has been improved by advice and testing by A Healey, N Walden, V Scarlett, R Walstead, S Carey, L Smith, J Vogel, J Willis, J Jenkins, T Juenger, and many others. Thanks to J Schnable, J Leebens-Mack, JG Monroe, CH Li, R Tarvin, and M Hufford for help refining the datasets and analyses presented in this manuscript. Thank you to Erich D Jarvis and the Vertebrate Genome Project members for advice and prepublication access to several genomes (budgerigar and dolphin). The work conducted by the US Department of Energy Joint Genome Institute is supported by the Office of Science of the US Department of Energy under Contract No. DE-AC02-05CH1123. Visualization was inspired in part by MCScanX and pairwise 'river' plots generated by other software. The use of syntenic orthogroups was originally inspired by work developed by CoGe; similar syntenic homology approaches have been implemented by other software, including pSONIC. MAW's work on this was supported by the National Institute of General Medical Sciences (NIGMS) of the National Institutes of Health (NIH) grant R35GM124827. JTL would like to thank Ashley Lovell, our friends and family for their support, which allowed him to work on this project during the difficult past 2 years.

## Additional information

### Funding

| Funder | Grant reference number | Author |
|---|---|---|
| U.S. Department of Energy | DE-AC02-05CH1123 | John T Lovell<br>Avinash Sreedasyam<br>Joseph W Carlson<br>David M Goodstein<br>Jeremy Schmutz |
| National Institute of General Medical Sciences | R35GM124827 | Melissa Wilson |

The funders had no role in study design, data collection, and interpretation, or the decision to submit the work for publication.

## Author contributions
John T Lovell, Conceptualization, Data curation, Software, Formal analysis, Visualization, Methodology, Writing – original draft, Project administration, Writing – review and editing; Avinash Sreedasyam, Conceptualization, Software, Formal analysis, Visualization, Methodology, Writing – original draft, Writing – review and editing; M Eric Schranz, Conceptualization, Formal analysis, Methodology, Writing – review and editing; Melissa Wilson, Formal analysis, Visualization, Methodology, Writing – original draft, Writing – review and editing; Joseph W Carlson, Data curation, Formal analysis, Methodology, Writing – original draft, Writing – review and editing; Alex Harkess, Conceptualization, Investigation, Writing – original draft, Writing – review and editing; David Emms, Software, Formal analysis, Methodology; David M Goodstein, Conceptualization, Software, Supervision, Funding acquisition, Visualization; Jeremy Schmutz, Conceptualization, Software, Funding acquisition, Methodology, Writing – original draft, Project administration, Writing – review and editing

## Author ORCIDs
John T Lovell ⓘ http://orcid.org/0000-0002-8938-1166
Avinash Sreedasyam ⓘ http://orcid.org/0000-0001-7336-7012
M Eric Schranz ⓘ http://orcid.org/0000-0001-6777-6565
Melissa Wilson ⓘ http://orcid.org/0000-0002-2614-0285
Alex Harkess ⓘ http://orcid.org/0000-0002-2035-0871
David Emms ⓘ http://orcid.org/0000-0002-9065-8978
David M Goodstein ⓘ http://orcid.org/0000-0001-6287-2697
Jeremy Schmutz ⓘ http://orcid.org/0000-0001-8062-9172

## Decision letter and Author response
Decision letter https://doi.org/10.7554/eLife.78526.sa1
Author response https://doi.org/10.7554/eLife.78526.sa2

# Additional files

## Supplementary files
• MDAR checklist

## Data availability
Raw data were sourced entirely from NCBI and Phytozome. Processed data, intermediate files, scripts, plots, and source data are all available in the data repository: https://github.com/jtlovell/GENESPACE_data (copy archived at swh:1:rev:77612f8c59fbfd43ef3f4c1719933bf0cbca3261). All source code and documentation for the GENESPACE R package can be found at https://github.com/jtlovell/GENESPACE (copy archieved at swh:1:rev:390341499ee1d2ccd5e1a894c4bd7c1bd20a3dda). An interactive viewer for the plant genomes can be found on phytozome at https://phytozome-next.jgi.doe.gov/tools/dotplot/synteny.html.

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
