## [Editor Report]

GENESPACE is a new and straightforward computational tool to include synteny information in the calculation of genome-wide sets of orthologs. The development of this tool is very timely as more and more complete chromosome-scale assembled genomes are becoming available. While the assembly problem has been solved, this is not the case for multiple genome comparisons, and GENESPACE is an important step to help remedy this gap in our comparative genomics toolbox.

---

## [Decision Letter]

**Decision letter after peer review:**

Thank you for submitting your article "GENESPACE: syntenic pan-genome annotations for eukaryotes" for consideration by *eLife*. Your article has been reviewed by 2 peer reviewers, one of whom is a member of our Board of Reviewing Editors, and the evaluation has been overseen by Detlef Weigel as the Senior Editor. The reviewers have opted to remain anonymous.

Essential revisions:

1) While you will see that the reviewers generally liked the functionality of GENESPACE and support its publication in *eLife*, there were multiple concerns regarding unclear descriptions of its methodology. It is therefore necessary that you revise the manuscript very carefully and make it accessible to both experts (including details on the methods and more formal definitions) and non-experts (on the general way how GENESPACE works).

*Reviewer #1 (Recommendations for the authors):*

While I support the publication of the manuscript, I have doubts that the manuscript in its current form is explaining well what GENESPACE is doing and how this is performed. While I do appreciate that the Results section tries to give an overview of the methodology, it is still very hard to follow as it is full of jargon and leaves out some important information, which only can be found in the methods.

How is synteny defined (line 105), which is the basis for dropping graph edges (which I assume are BLAST hits) before running OrthoFinder (while in Figure S1 orthofinder is run before the synteny module)? Once the BLAST hits between non-syntenic regions are deleted OrthoFinder can be run without any adjustments, but still connect genes to OrthoGroup in syntenic regions only. However, the sentence in line 108 is not clear, how can BLAST hits in syntenic regions be merged with synteny-constrained OGs (where BLAST hits in syntenic regions are actually the basis for synteny-constrained OGs)? In this regard, it is not clear what the differences between synteny-constrained OGs and within-block OGs are.

Table 1. It is not clear what this table actually shows and requires guessing. (Why is this analysis done with nine chromosomes only?). It becomes more clear after rereading the later section that explains Table 1, but on the first occurrence of table 1 in the text, I got lost. The GENESPACE run using the tetraploid genome still relies on the synteny between the subgenomes in order to generate OG groups of size of 6? It is not fully clear what the differences between the tetraploid run and the split by subg. runs are when synteny information between the subgenomes is also needed in the tetraploid run. Besides the description of this approach is very hard to follow. I also do not see why this approach reveals sensitivity – for example in the pure OrthoFinder runs some of the OGs of size of 6 might be extended by in-paralogs which is (as far as I understand) the goal of OrthoFinder.

In a second paragraph, GENESPACE is also presented as a tool for the identification of syntenic regions (which presumably are the syntenic regions that the OG analysis are based on, however, this is not clear). GENESPACE goal is to filter BLAST hits in a way that regions are linked to only a single region in the other genome by "subsetting the BLAST hits to those 181 within the same orthogroups". This again is not clear, does this imply that OrthoFinder results are used to filter the BLAST hits? The blast hits are then extended. Though there is more about this in the methods, it is not clear to me, why this leads to non-overlapping syntenic regions, specifically, as connected orthologs do not need to be right next to each other.

I think the expression "concept of a pan-genome annotation" or just "pan-genome annotation" in itself is misleading. GENESPACE is a tool for ortholog identification and not for gene annotation. I see the point why it is called like this (all genes are projected on one reference genome), however, I actually expected that GENESPACE would annotate/correct gene structure annotations, which is the typical context of the term "annotation" in genomics.

It would be helpful to define (the differences of) synteny and collinearity (L49 and rest of manuscript).

How is the exact location defined when projecting non-reference OG to reference positions?

In multiple parts of the text, it is mentioned that tandem arrays are excluded or treated in a different way. Could this be clarified?

It is not clear how the "second proximity search step" in GENESPACE actually works. Where are the significant gains in single copy genes coming from? Why are they not already identified?

*Reviewer #2 (Recommendations for the authors):*

The code and data are available on GitHub.

The algorithm could be described more clearly. For this, it would probably help to introduce key definitions more formally. Below is a list of questions I was left with after reading the manuscript.

When you use 'syntenic regions', are regions in multiple genomes meant or always only in two genomes? If appropriate you could qualify the term according to the usage.

The riparian plot appears to use only n-1 pairs of genomes out of the n*(n-1) pairs of different genomes for which syntenic regions were computed. What is the formal criterion or algorithm for that choice that was made? The respective comments in the Figure captions are not clear to me.

What does GENESPACE do if a gene or region is missing in reference genome?

How are the syntenic regions extracted and how is can this be adjusted by the user (line 620)?

I do not understand how 'collinear arrays' are defined (line 635). What does it mean for a group of genes to share an orthogroup?

How is the step done from pairwise synteny to synteny of multiple genomes? Is this done at all? The manuscript reads around line 108 as if OrthoFinder is (re)run on syntenic regions. This makes sense if there are in general multiple (>2) regions considered to be mutually syntenic with one another. If that is the case, how do you address the choice of having a few long syntenic regions versus many short syntenic regions when both are plausible and the choices share a subset of genes?

Can there be two genes in an orthogroup that GENESPACE outputs that are not syntenic within their pair of genomes?

Does GENESPACE scale quadratically with the number of genomes?

Why are for two genomes g1 and g2 both (g1, g2) and (g2, g1) contained, e.g. in Supplementary File 1?

(Why) is it not symmetric? What are the roles of first and second genomes?

Table 1: The row labels are hard to make sense of. Are any rows the results of GENESPACE?

Figure 1: The human Y chromosome is in the figure caption but not in the image. The same holds for Supplementary Figure 3.

OrthoFinder is inconsistently spelled as OrthoFinder, Orthofinder and orthofinder.

Description of Supplemental Data 1. There are no columns pgChr or pgOrd in Lovell_09-03-2022-TR-*eLife*-78526_Supplementary_File_1.txt

Apparently the order of files and descriptions is not consistent with each other.

Line items:

43: grammar

162: What does 'contrasted' mean here?

173: What does 'dosage' mean here?

713: Add 'of dotplots' as the link apparently does not contain all of GENESPACEs results.

[Editors’ note: further revisions were suggested prior to acceptance, as described below.]

Thank you for resubmitting your work entitled "GENESPACE: syntenic pan-genome annotations for eukaryotes" for further consideration by *eLife*. Your revised article has been evaluated by Detlef Weigel (Senior Editor) and a Reviewing Editor.

The manuscript has been improved but there are some remaining issues that need to be addressed, as outlined below:

*Reviewer #1 (Recommendations for the authors):*

The authors have addressed many of my concerns by rewriting large parts of the manuscript, which significantly improved the description of what GENESPACE does and how it works. In particular the step-by-step descriptions in the methods worked well for me. I fully support publication of this manuscript, though there are some small suggestions below.

In the first Results section, I would have found it helpful to read one sentence on what OrthoFinder and MCScanX do (resp. what they output).

Line 71: This sentence was still not clear to me. What is the gene rank that is recalculated, and why would a recalculation make genes with PAV?

Table 2: no horizontal bars visible for me as indicated in the legend.

*Reviewer #2 (Recommendations for the authors):*

In the revision, the authors have made substantial changes. In particular the first two tables were removed from the original manuscript. Lovell et al., have added a new Table 1, in which they compare their tool GENESPACE to OrthoFinder and MCScanX on pairs of genome annotations, and an in-depth description of their implementation steps.

The example plots are convincing, not so the benchmark in its current form. I am missing a measure of sensitivity or coverage as well as a measure of how much the orthogroups respect synteny.

In the very important sentence in line 87 it is grammatically ambiguous to what ‘that’ refers to. It should be reformulated so that it is unambiguous even for readers not understanding the authors intention yet.

I assume ‘that’ in line 87 qualifies ‘orthogroup’. Under that assumption, a trivial and very insensitive bogus orthology finder that outputs in a genome-wide search for synteny a single orthogroup with (any) two genes in two different species would achieve 2/2=100% of what the authors refer to as "accuracy and precision". A little less bogus, a trivial filter method that randomly (and independent of synteny) reduces an orthogroup with multiple genes from different chromosomes of the same species to one with genes on at most one chromosome per species would perform better in this measure than its orthogroup input. The numbers of Table 1 can consequently by themselves not reasonably establish the ”outperformance” claimed in line 89.

Line 88 appears to refer to the same results of Table 1 but formulates “percentage of … orthogroups” as opposed to “percent of genes”, which is of course different. I strongly suggest the authors use a precise definition of their central measures of accuracy, e.g. with a formula in which all terms are themselves unambiguously defined.

The methods are now described at length and include coding details on pipelining, thresholding and implementation flow. Admittedly, this reviewer had requested details on thelgorithmm, but had ideally expected a concise and precise formulation of the formal objectives that GENESPACE achieves. Most readers would arguably appreciate a more succinct formulation that is specific about where the main ideas of the program lie and to which high-level design decisions are made. For example, is the decision which homologous genes are syntenic based on all-versus-all pairwise comparisons of coordinates, on pairwise comparisons of neighboring species in a user-specified order, in an order chosen by the program, are comparisons only done all-against-reference or even are comparisons between two genomes informed by third genomes?

---

## [Author Response]

Essential Revisions (for the authors):1) While you will see that the reviewers generally liked the functionality of GENESPACE and support its publication in eLife, there were multiple concerns regarding unclear descriptions of its methodology. It is therefore necessary that you revise the manuscript very carefully and make it accessible to both experts (including details on the methods and more formal definitions) and non-experts (on the general way how GENESPACE works).

We have expanded the methods and worked to make descriptions clearer. See line-specific comments below.

Reviewer #1 (Recommendations for the authors):While I support the publication of the manuscript, I have doubts that the manuscript in its current form is explaining well what GENESPACE is doing and how this is performed. While I do appreciate that the Results section tries to give an overview of the methodology, it is still very hard to follow as it is full of jargon and leaves out some important information, which only can be found in the methods.

We fully agree. The methodological descriptions in the results were simultaneously too in-theweeds/jargony and not detailed enough. We opted to address this issue with three big changes: (1) we add box 1 and Figure 1, which provide technical definitions and a visual description of how GENESPACE works, respectively; (2) we re-framed the first section of the results to provide less technical detail but more explicit descriptions of how the pipeline works at a high level, with references to processes detailed in Figure 1; and (3) we restructured the methods to first provide an initial high-level overview of the pipeline at the beginning of the methods section then a complete description of each step in the pipeline later in the methods.

How is synteny defined (line 105), which is the basis for dropping graph edges (which I assume are BLAST hits) before running OrthoFinder (while in Figure S1 orthofinder is run before the synteny module)?

This is an example of jargon that we have dropped. Now the methods are described as blast hits and not in graph terms (except where unavoidable), which hopefully makes it easier to follow. The methods to infer synteny are now fully laid out in Figure 1 and the methods [345-428].

Once the BLAST hits between non-syntenic regions are deleted OrthoFinder can be run without any adjustments, but still connect genes to OrthoGroup in syntenic regions only. However, the sentence in line 108 is not clear, how can BLAST hits in syntenic regions be merged with synteny-constrained Ogs (where BLAST hits in syntenic regions are actually the basis for synteny-constrained Ogs)? In this regard, it is not clear what the differences between synteny-constrained Ogs and within-block Ogs are.

Yes, another case of opaque methods in the previous version. We have hopefully improved this description [331-344]. This step has also been added to Figure 1.

Table 1. It is not clear what this table actually shows and requires guessing. (Why is this analysis done with nine chromosomes only?). It becomes more clear after rereading the later section that explains Table 1, but on the first occurrence of table 1 in the text, I got lost. The GENESPACE run using the tetraploid genome still relies on the synteny between the subgenomes in order to generate OG groups of size of 6? It is not fully clear what the differences between the tetraploid run and the split by subg. Runs are when synteny information between the subgenomes is also needed in the tetraploid run. Besides the description of this approach is very hard to follow. I also do not see why this approach reveals sensitivity – for example in the pure OrthoFinder runs some of the Ogs of size of 6 might be extended by in-paralogs which is (as far as I understand) the goal of OrthoFinder.

Thanks for this – we agree that the cotton subgenome comparison is not the best example. We have now revisited the sensitivity/precision estimates for GENESPACE and have followed reviewer #2’s advice and changed our estimates of sensitivity to pairs of genomes across a range of phylogenetic distances [81108]. We hope that these 1:1 comparisons are easier to follow. The confusing cotton-subgenome analyses have been dropped.

In a second paragraph, GENESPACE is also presented as a tool for the identification of syntenic regions (which presumably are the syntenic regions that the OG analysis are based on, however, this is not clear). GENESPACE goal is to filter BLAST hits in a way that regions are linked to only a single region in the other genome by “subsetting the BLAST hits to those 181 within the same orthogroups”. This again is not clear, does this imply that OrthoFinder results are used to filter the BLAST hits? The blast hits are then extended. Though there is more about this in the methods, it is not clear to me, why this leads to non-overlapping syntenic regions, specifically, as connected orthologs do not need to be right next to each other.

We have now directly addressed this with a specific example of how GENESPACE improves syntenic blocks in Figure 1. We hope this is clearer than the cotton comparisons.

I think the expression “concept of a pan-genome annotation” or just “pan-genome annotation” in itself is misleading. GENESPACE is a tool for ortholog identification and not for gene annotation. I see the point why it is called like this (all genes are projected on one reference genome), however, I actually expected that GENESPACE would annotate/correct gene structure annotations, which is the typical context of the term “annotation” in genomics.

We feel that “pan-genome annotation” is the logical extension: a genome annotation is the set of sequences that form genes in one genome, so a pan-genome annotation would be the set of sequences that form genes across multiple genomes. However, we understand the reviewer’s point and now provide an explicit definition up front. We do have scripts to do the gene model improvement/pseudogene finding that the reviewer suggests. We envision integrating these methods with GENESPACE in the next year or two.

It would be helpful to define (the differences of) synteny and collinearity (L49 and rest of manuscript).

We have added a definitions box that presents this distinction.

How is the exact location defined when projecting non-reference OG to reference positions?

We have hopefully dropped confusing text in the results / introduction. We have also provided extensive details about how this process works in the methods [438-463].

In multiple parts of the text, it is mentioned that tandem arrays are excluded or treated in a different way. Could this be clarified?

Yes, this was confusing. We have now addressed this directly in the main text, including a definition in box1, and expanded the description in the methods [364-370]. The example of the pan-genome annotation in Figure 1 also hopefully helps.

It is not clear how the “second proximity search step” in GENESPACE actually works.

We have excluded this complexity from the pipeline description in the main text and now go into far more detail in the methods [424-428, 499-509].

Where are the significant gains in single copy genes coming from? Why are they not already identified?

We now describe the problem/solution in more detail at the beginning of the Results section [62-78]. The main difference can be observed in Figure 1, showing the difference between a single-step MCScanX and GENESPACE.

Reviewer #2 (Recommendations for the authors):The code and data are available on GitHub.The algorithm could be described more clearly. For this, it would probably help to introduce key definitions more formally. Below is a list of questions I was left with after reading the manuscript.

We have added a definitions box and provided a much expanded description of the pipeline in the methods.

When you use ‘syntenic regions’, are regions in multiple genomes meant or always only in two genomes? If appropriate you could qualify the term according to the usage.

This is a good question – synteny itself is always a pairwise pattern, but one that can be chained among multiple genomes. We have clarified this distinction when explaining the methods [354-356].

The riparian plot appears to use only n-1 pairs of genomes out of the n*(n-1) pairs of different genomes for which syntenic regions were computed. What is the formal criterion or algorithm for that choice that was made? The respective comments in the Figure captions are not clear to me.

*We are not sure we follow this question – here, we reply to the question: “how does GENESPACE choose the chained order of genomes in the riparian plot”. If this isn’t right, please clarify and we are happy to respond in kind*. The riparian plot genome order is a query of all possible pairwise combinations to the chain specified by the user in the genomeIDs parameter (default is the order specified in the init_genespace step). The user can apply any algorithm or criteria they wish to order the genomes. For example, one option would be a TSP solver path to minimize the number of inversions. We have hopefully clarified the figure captions [585-586, 592]

What does GENESPACE do if a gene or region is missing in reference genome?

In terms of the synteny step: PAV is invisible in pairwise synteny methods since we deal with gene rankorder position that is collapsed to genes with solid (usually orthogroup-constrained) hits between the two genomes. That is, if genes 1-3 and 8-10 have direct hits between genome A and B, but genes 4-5 have no hit in genome B and 6-7 have no hits in genome A, GENESPACE synteny() would only see genes 1-3,810 ordered 1,2,3,4,5,6. We now describe this at a high level in Figure 1 and in detail in the methods [390391, 400-405].

In terms of the pangenome step: the syntenic position of all genes (within syntenic block breakpoints) are interpolated against the reference (see new detailed methods [438-463]). For a syntenic orthogroup that lacks a reference genome gene, the median interpolated position is used. If multiple high-confidence interpolated positions exist, that orthogroup is given multiple positions (e.g. in a polyploid).

How are the syntenic regions extracted and how is can this be adjusted by the user (line 620)?

We now provide a full description of the synteny pipeline in the methods [345-428] and specifically address syntenic region extraction from raw blast hits to output.

I do not understand how ‘collinear arrays’ are defined (line 635). What does it mean for a group of genes to share an orthogroup?

This was confusing and hopefully has been remedied by using the existing term “tandem” instead of collinear. While a physically proximate group of genes in the same orthogroup (aka “share an orthogroup” … now changed) is not necessarily a tandem array, all true tandem arrays will fall into this category, so we feel the definition is ok, considering that tandem is a far more well understood term.

How is the step done from pairwise synteny to synteny of multiple genomes? Is this done at all? The manuscript reads around line 108 as if OrthoFinder is (re)run on syntenic regions. This makes sense if there are in general multiple (>2) regions considered to be mutually syntenic with one another. If that is the case, how do you address the choice of having a few long syntenic regions versus many short syntenic regions when both are plausible and the choices share a subset of genes?

All steps related to synteny are done in pairs. Overall synteny is never considered and only concatenated into the pan-genome annotation. This step lets a user query the syntenic region across any number of genomes.

Can there be two genes in an orthogroup that GENESPACE outputs that are not syntenic within their pair of genomes?

Non-syntenic orthologs are added to the pan-genome (see Figure 1 for examples and methods [464-482]). All hits, regardless of whether they are syntenic or in the same orthogroup are returned as well.

Does GENESPACE scale quadratically with the number of genomes?

Default OrthoFinder scales quadratically (it does all v. all blast). GENESPACE and orthofinderMethod = “fast” scales n+(n choose 2) since pairwise synteny is only inferred in one direction.

Why are for two genomes g1 and g2 both (g1, g2) and (g2, g1) contained, e.g. in Supplementary File 1?

I wonder if there was an issue with naming/uploading of the SI data in the previous submission. It looks good on my end, but here and below, there is some confusion – SI data 1 *should* be the pan-genome annotation for the vertebrates against the human coordinate system. I think the reviewer is probably referring to SI data 3, 4 etc which are syntenic block breakpoints and should be a long-formatted symmetric table. We now do not upload the SI as individual files but as a single zip archive with the file names retained. That way there should be no confusion.

(Why) is it not symmetric?

I have checked and block breakpoint tables are coercible to symmetric arrays [genome1, genome2, block].

What are the roles of first and second genomes?

This is for backwards compatibility with the phytozome visualization tool. Plus, it lets the user query by their genomes of interest without having to know which served as the target and which the query.

Table 1: The row labels are hard to make sense of. Are any rows the results of GENESPACE?

Dropped.

Figure 1: The human Y chromosome is in the figure caption but not in the image. The same holds for Supplementary Figure 3.

Good catch. There is no syntenic to the human Y. We have dropped it from the captions.

OrthoFinder is inconsistently spelled as OrthoFinder, Orthofinder and orthofinder.

Fixed to the correct name throughout.

Description of Supplemental Data 1. There are no columns pgChr or pgOrd in Lovell_09-03-2022-TR-eLife-78526_Supplementary_File_1.txtApparently the order of files and descriptions is not consistent with each other.

Yes, there was something broken in how SI tables were specified (see above). We have attempted to fix this by just uploading a single.zip archive with all SI data.

Line items:43: grammar162: What does ‘contrasted’ mean here?173: What does ‘dosage’ mean here?713: Add ‘of dotplots’ as the link apparently does not contain all of GENESPACEs results.

These have been fixed

[Editors' note: further revisions were suggested prior to acceptance, as described below.]

Reviewer #1 (Recommendations for the authors):The authors have addressed many of my concerns by rewriting large parts of the manuscript, which significantly improved the description of what GENESPACE does and how it works. In particular the step-by-step descriptions in the methods worked well for me. I fully support publication of this manuscript, though there are some small suggestions below.In the first Results section, I would have found it helpful to read one sentence on what OrthoFinder and MCScanX do (resp. what they output).

We have re-worked the intros to synteny and orthology to include references to these programs [65-67].

Line 71: This sentence was still not clear to me. What is the gene rank that is recalculated, and why would a recalculation make genes with PAV?

This wasn’t immediately obvious. We have updated the description to hopefully be clearer [76-81].

Table 2: no horizontal bars visible for me as indicated in the legend.

We have removed this statement as it referred to an earlier version of the table. Thanks for the catch.

Reviewer #2 (Recommendations for the authors):In the revision, the authors have made substantial changes. In particular the first two tables were removed from the original manuscript. Lovell et al., have added a new Table 1, in which they compare their tool GENESPACE to OrthoFinder and MCScanX on pairs of genome annotations, and an in-depth description of their implementation steps.The example plots are convincing, not so the benchmark in its current form. I am missing a measure of sensitivity or coverage as well as a measure of how much the orthogroups respect synteny.

We are interpreting this as “how many orthologous gene pairs are not syntenic”. GENESPACE provides these as flagged entries in the pan-genome annotation. We mentioned this in the methods and include examples in Figure 1. We have now included statistics about the number of non-syntenic orthologs in a few comparisons [112-119]. Since GENESPACE provides the non-syntenic orthologs to the user, their presence is not problematic and can be interpreted however the user sees fit.

In the very important sentence in line 87 it is grammatically ambiguous to what 'that' refers to. It should be reformulated so that it is unambiguous even for readers not understanding the authors intention yet.

The sentence in line 87 reads: “To estimate accuracy and precision of each approach, we calculated the percent of genes in orthogroups and syntenic blocks that were placed on exactly one chromosome per genome (Table 1).” It seems clear to us that “that” refers to both orthogroups and syntenic blocks, especially since table 1 shows both these attributes. Nonetheless, we have modified this sentence slightly and added (a) and (b) to the table to remove any ambiguity [96-97].

I assume 'that' in line 87 qualifies 'orthogroup'. Under that assumption, a trivial and very insensitive bogus orthology finder that outputs in a genome-wide search for synteny a single orthogroup with (any) two genes in two different species would achieve 2/2=100% of what the authors refer to as "accuracy and precision". A little less bogus, a trivial filter method that randomly (and independent of synteny) reduces an orthogroup with multiple genes from different chromosomes of the same species to one with genes on at most one chromosome per species would perform better in this measure than its orthogroup input. The numbers of Table 1 can consequently by themselves not reasonably establish the "outperformance" claimed in line 89.

This example would be appropriate if GENESPACE made orthogroups de novo. However, it does not. We have clarified this [81]; GENESPACE uses OrthoFinder orthogroups as the starting point. Since comparisons in Table 1 are made solely between the GENESPACE-refined orthogroups and raw OrthoFinder orthogroups, measuring the proportion of orthogroups that match the expected evolutionary patterns is the most obvious and non-trivial measure we can define.

Line 88 appears to refer to the same results of Table 1 but formulates "percentage of … orthogroups" as opposed to "percent of genes", which is of course different. I strongly suggest the authors use a precise definition of their central measures of accuracy, e.g. with a formula in which all terms are themselves unambiguously defined.

We feel that a formal formula is overkill, since it is a simple statistic that we define clearly in the caption: “We present the percent of genes that were found in an orthogroup that hit a single chromosome per genome…”. We have altered the text to clarify that the values are in terms of genes, not orthogroups [98-99]. Thanks for this catch.

The methods are now described at length and include coding details on pipelining, thresholding and implementation flow. Admittedly, this reviewer had requested details on the algorithm, but had ideally expected a concise and precise formulation of the formal objectives that GENESPACE achieves. Most readers would arguably appreciate a more succinct formulation that is specific about where the main ideas of the program lie and to which high-level design decisions are made. For example, is the decision which homologous genes are syntenic based on all-versus-all pairwise comparisons of coordinates, on pairwise comparisons of neighboring species in a user-specified order, in an order chosen by the program, are comparisons only done all-against-reference or even are comparisons between two genomes informed by third genomes?

In the revision, we significantly expanded both the overall description (as previously suggested by reviewer 1) and pipeline details (previously suggested by reviewer 2). The high-level decisions are all outlined in the new figure 1 and the formal objectives both in the second paragraph of the results [72-82] and second paragraph of the methods [280-288]. The example reviewer 2 provides here represents a pipeline detail that is thoroughly described in the methods [e.g. 364-365].